# ONE SIZE DOES NOT FIT ALL: CROSS-ARCHITECTURAL LAYER-WISE REPRESENTATIONS IN DIFFUSION MODELS

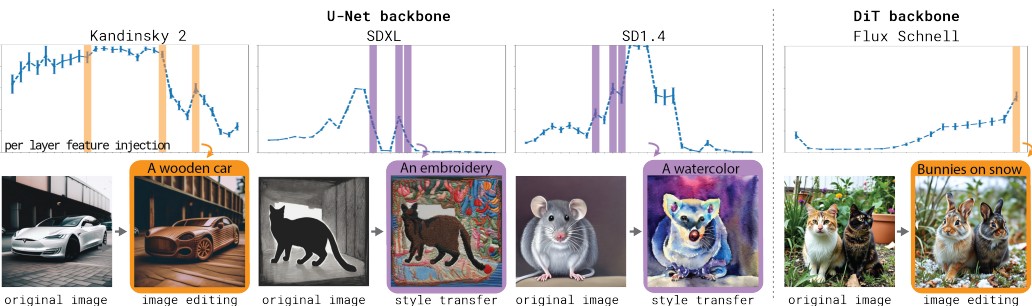

Figure 1: Layer-wise feature injection across diffusion architectures. (Top) plots showing SSIM similarity curves from the original (O) to the edited (E) image of layerwise feature injection, orange highlight indicates layers where semantic mixing occurs. Orange layers are used for editing (highest SSIM). Purple for style transfer (second highest) . (Bottom) original image and edited image, editing prompt over the edited image.

## ABSTRACT

Recent advances in diffusion models have enabled powerful text-to-image synthesis and training-free editing. However, despite growing architectural diversity, most editing techniques rely on implicit assumptions about shared internal representations across models. In this paper, we conduct a systematic, layer-wise analysis of internal representations across a wide range of diffusion architectures, including Stable Diffusion (SD1.4, SD2, SDXL), Kandinsky, and DiT-based models (SD3.5, Flux). We quantify how semantic and stylistic information propagates through U-Net backbones and their transformer-based counterparts using a targeted feature injection protocol. Our findings uncover architecture-specific encoding patterns, such as symmetric representational flow in SD1.4/2.0, bottleneck centrality in SDXL, decoder-centric representation in Kandinsky, and middle-late semantic representation formation in DiTs. We further show that adversarially distilled models preserve, but amplify, their teacher's representational structure. These insights inform a principled injection-based framework for text-guided image editing and style transfer. To the best of our knowledge, we are the first to achieve successful editing on such a broad range of models.

## 1 INTRODUCTION

Diffusion models have emerged as a powerful family of generative models, enabling high-quality text-to-image synthesis across diverse domains. While their expressive capacity is remarkable, understanding and controlling their internal representations remains an open challenge. A growing line of work attempts training-free editing by leveraging *feature injection*, i.e., transferring activations between the denoising streams of an original and an editing prompt (Hertz et al.; Tumanyan et al., 2023; Liu et al., 2024; Jeong et al., 2024b; Jiang & Chen, 2024; He et al., 2024). These methods demonstrate strong results but rely on an implicit assumption: diffusion backbones share similar representational structures.

Table 1: Overview of image editing, style transfer, and personalization methods. Each column marks the modules used (✓) or not used (–) for injection together with the layers injected, model(s), and reference. As evidenced in the Model column, **existing methods are explicitly tested only on one or, at most, two models**.

| Method | Model | Modules | | | | | Layers |
|---|---|---|---|---|---|---|---|
| | | SelfAtt | CrossAtt | ResNet | Skip | Text enc. | |
| **Image editing** | | | | | | | |
| SDEdit (Meng et al.) | DDPM | – | – | – | – | – | — |
| Prompt-to-Prompt (Hertz et al.) | Imagen | ✓ | ✓ | – | – | – | — |
| Plug-and-Play (Tumanyan et al., 2023) | SD 1.4–2 | ✓ | – | ✓ | – | – | Layer 4 |
| DiffQuickFix (Basu et al.) | SD 1.4 | ✓ | ✓ | ✓ | – | ✓ | — |
| PromptFree Editing (Liu et al., 2024) | SD 1.5 | – | ✓ | – | – | ✓ | Layers 4–14 |
| **Style transfer** | | | | | | | |
| InjectFusion (Jeong et al., 2024a) | DDPM | – | – | ✓ | – | – | Bottleneck |
| Artist (Jiang & Chen, 2024) | SD 2.1 | ✓ | – | ✓ | – | – | Content: 4–6, Style: 4–12 |
| FreeStyle (He et al., 2024) | SD XL | – | – | – | ✓ | – | — |
| Z* (Deng et al., 2024) | SD 1.5 | – | ✓ | – | – | – | Decoder |
| **Personalization** | | | | | | | |
| P+ (Voynov et al., 2023) | SD 1.4 | – | ✓ | – | – | ✓ | Content: 5-7, Style: 1-4, 8-12 |
| ProSpect (Zhang et al., 2023) | SD 1.4 | – | ✓ | – | – | ✓ | — |
| MATTE (Agarwal et al., 2023) | SD | – | ✓ | – | – | ✓ | Content: 5-7, Color: 1-4, 8-12 |

This assumption is increasingly problematic as architectures diversify. U-Net variants (e.g., Stable Diffusion 1.x, 2.x, XL (Rombach et al., 2022), Kandinsky (Razzhigaev et al., 2023)) differ in how they distribute attention, use decoders to image space, and organize residual connections, while recent Diffusion Transformers (DiTs, (Peebles & Xie, 2023)) adopt entirely different representational dynamics. Yet, as shown in Table 1, most existing methods are only validated on a single model, raising the question of whether their effectiveness is due to universal principles or architecture-specific artifacts.

Understanding whether representational roles are consistent across models is crucial for at least three reasons: (i) it advances mechanistic interpretability by revealing where semantic and stylistic information is encoded, (ii) it informs architectural choices by highlighting inductive biases of different designs, and (iii) it provides a principled foundation for generalizable editing and control methods.

In this paper, we conduct the first systematic, layer-wise analysis of internal representations across multiple diffusion backbones. We probe how semantic and stylistic information propagates through models, including Stable Diffusion 1.4, 2, XL, 3.5 (Rombach et al., 2022; Podell et al.; Esser et al., 2024), Kandinsky (Razzhigaev et al., 2023), Flux (Labs, 2024), and turbo-distilled variants (Sauer et al., 2024a). Our methodology sequentially injects activations from the original denoising process into the editing stream and evaluates the resulting images along structural (SSIM, keypoints) and stylistic (color, texture) dimensions. This probing framework reveals substantial differences in representational organization: e.g., SD1.4 and SD2 exhibit symmetric flows through encoder–decoder blocks, SDXL centralizes information in its bottleneck, Kandinsky shifts representational burden to decoder layers, while DiT-based models concentrate on middle-late transformer blocks.

Building on these insights, we introduce a principled injection-based pipeline for text-driven image editing and style transfer that adapts layer selection to each architecture. To the best of our knowledge, this is the first method to perform image editing and style transfer that can be applied to such a variety of models. We further show that this paradigm generalizes beyond U-Nets to DiT-based diffusion models (Figure 1). Our findings challenge the assumption of uniformity across backbones, highlight architecture-specific encoding strategies, and provide both methodological and conceptual tools for advancing controllability and interpretability in diffusion models.

**Contributions.** Our main contributions are threefold: (i) We present the first systematic, layer-wise injection-based probing of internal representations across a wide range of diffusion backbones, revealing that semantic and stylistic information is encoded in architecture-specific ways. (ii) We introduce a probing-based methodology that quantifies the representational role of each layer using structural and stylistic similarity metrics. (iii) Leveraging these insights, we design an injection-

based pipeline for text-driven image editing and style transfer, and demonstrate its generalization across both U-Net- and DiT-based diffusion models.

## 2 RELATED WORK

Current methods for training-free editing largely rely on *feature injection* (Hertz et al.; Liu et al., 2024; Tumanyan et al., 2023; Jiang & Chen, 2024; He et al., 2024). These approaches run two parallel denoising processes, one conditioned on the original image and one on the editing prompt, and achieve editing by injecting activations from the former into the latter. The effectiveness of such methods depends critically on the choice of modules, layers, and timesteps of injection, often combined with modulation mechanisms that regulate the strength of the transfer (Table 1). In this sense, feature injection provides a natural probe of the representational structure of diffusion backbones.

Much of the literature has emphasized the role of attention modules. Self-attention has been linked to spatial affinity propagation, while cross-attention has been associated with object layout and geometry (Hertz et al.; Liu et al., 2024; Tumanyan et al., 2023). Yet, findings diverge: for example, DiffQuickFix (Basu et al.) reports that most cross-attention layers do not causally affect outputs, whereas FreePromptEdit (Liu et al., 2024) shows that they carry object-level information. By contrast, residual blocks and skip connections have received less attention, despite evidence (e.g., Skip-Inject (Schaerf et al., 2025)) that late encoder skip connections transmit crucial spatial information. Similarly, the representational roles of the bottleneck and deeper decoder layers remain debated: some works highlight their importance for content and object information (Jeong et al., 2024b; Haas et al., 2024), while others stress a coarse division between shallow, style-oriented layers and deeper, content-oriented ones (Voynov et al., 2023; Agarwal et al., 2023).

While U-Nets reveal systematic representational patterns, it remains unclear whether these *generalize across architectures*. Prior analyses typically focus on a single model, leaving open whether observed behaviors reflect universal principles of diffusion backbones or model-specific design choices. Clarifying this distinction is important, as convergence would suggest shared principles that enable transferable editing and control strategies, whereas divergence would imply the need for backbone-specific approaches. We address this gap by taking a cross-architectural view, probing layer- and module-level behaviors in the injection paradigm.

## 3 PRELIMINARIES

This section provides a brief overview of the diffusion models considered in our analysis. We focus on architectural differences, which influence internal representations, and we introduce the notation used throughout the paper.

### 3.1 MODELS

We analyze models based on two types of backbone architectures: U-Nets and Diffusion Transformers (DiTs). U-Net-based backbones in consideration share three components: (i) an encoder–decoder operating in a reduced noise space, (ii) attention modules integrated into intermediate blocks, and (iii) a text encoder for conditioning. DiTs replace the U-Net with a ViT (Dosovitskiy et al., 2020) with several stacked transformer blocks. A general description of latent diffusion models (LDMs) (Rombach et al., 2022) and the U-Net backbone (Ronneberger et al., 2015) is deferred to the Appendix 7.

These models are selected among the most widely adopted to cover a broad range of backbones (DiT-U-Net, but also discrete-continuous decoder), as well as cases with minor but known differences (i.e., almost the same model trained on different datasets, removal of a layer, different density of attention modules). We analyze the following models:

- **Stable Diffusion 1.4** (Rombach et al., 2022): four encoder and four decoder layers, each with three attention modules. Text conditioning uses OpenAI's CLIP (Radford et al., 2021), and outputs are decoded via a lightweight Variational Autoencoder (VAE) (Kingma & Welling, 2013).
- **Stable Diffusion 2**: identical to SD1.4, except trained on LAION-5B (Schuhmann et al., 2022) with OpenCLIP for conditioning, resulting in a larger text embedding space (768 vs. 512).

Table 2: Summary table of the models. CLIP here refers to OpenAI's CLIP.

| Model | Backbone | Text conditioning | Decoder | Training | Distillation |
|---|---|---|---|---|---|
| SD 1.4 | U-Net | CLIP | VAE | LAION-5B | - |
| SD 2 | U-Net | OpenCLIP | VAE | LAION-5B (w/o NFSW) | - |
| SD 2 turbo | U-Net | OpenCLIP | VAE | - | ✓ |
| SD XL | U-Net | CLIP + OpenCLIP | VAE | NA | - |
| SD XL turbo | U-Net | CLIP + OpenCLIP | VAE | - | ✓ |
| Kandinsky | U-Net | OpenCLIP | MoVQGAN | LAION HD | - |
| SD 3.5 turbo | DiT | CLIP + OpenCLIP + T5 | VAE | - | ✓ |
| Flux Schnell | DiT | T5 | VAE | - | ✓ |

- **Stable Diffusion XL** (Podell et al.): reduces the number of encoder/decoder layers, concentrating attention in deeper layers. It conditions jointly on OpenAI CLIP and OpenCLIP and is paired with a refiner network for high-frequency detail.
- **Kandinsky** (Razzhigaev et al., 2023): departs more strongly from SD. It replaces the VAE decoder with MoVQGAN (Zheng et al., 2022), integrates an image prior to map text embeddings into CLIP image space (similar to unCLIP (Ramesh et al., 2022)), and supports multilingual prompts.
- **Turbo Variants** (Sauer et al., 2024a): adversarially distilled versions of SD2 and SDXL that enable generation in a handful of inference steps.
- **Stable Diffusion 3.5 Turbo** (Esser et al., 2024): a multimodal diffusion transformer distilled via Adversarial Diffusion Distillation (ADD) (Sauer et al., 2024b). It integrates three fixed pretrained text encoders (OpenCLIP-ViT/G, CLIP-ViT/L, T5-XXL), employs Query–Key normalization for stability, and comprises 38 transformer layers.
- **Flux Schnell** (Labs, 2024): a rectified-flow hybrid transformer, distilled with Latent Adversarial Diffusion Distillation (LADD) (Sauer et al., 2024a), achieving high-quality outputs in 1–4 steps. Its architecture includes 19 transformer blocks and 39 additional single transformer blocks.

A summary of the models and their main characteristics is provided in Table 2.

## 3.2 NOTATION

Following the naming convention of diffusers, we denote U-Net encoder blocks as `down` (3–4 groups of 2–3 layers), the bottleneck as `mid`, and decoder blocks as `up`. Residual layers are referred to as `resnets`, and attention outputs as `attentions`. Throughout, we use the term *layer* to denote the output of a single functional block (convolution, transformer, or residual unit). A schematic of the U-Net architecture with these notations is shown in Figure 2.

## 4 ONE SIZE DOES NOT FILL ALL

Next, we show that diffusion backbones lack a one-size-fits-all representational structure; instead, each architecture exhibits distinct internal structure. We identify these by performing layer-wise feature injection and quantifying image similarity across structural and stylistic metrics.

**Feature Injection.** To isolate the influence of individual layers, we perform targeted *feature injection*. For a given denoising step

$$z_{t-1} = f(z_t) = f_L \circ f_{L-1} \circ \cdots \circ f_{i+1} \circ f_i \circ f_{i-1} \circ \cdots \circ f_1(z_t), \tag{1}$$

where $f$ denotes U-Net layers and $\circ$ composition, we replace the activations at layer $i$ from the editing trajectory $E$ with those from the original image trajectory $O$:

$$f_l = \begin{cases} f_l^O, & \text{if } l = i, \\ f_l^E, & \text{otherwise.} \end{cases} \tag{2}$$

This procedure allows us to attribute downstream changes to the representational contribution of layer $i$. This setup is schematized in Figure 2.

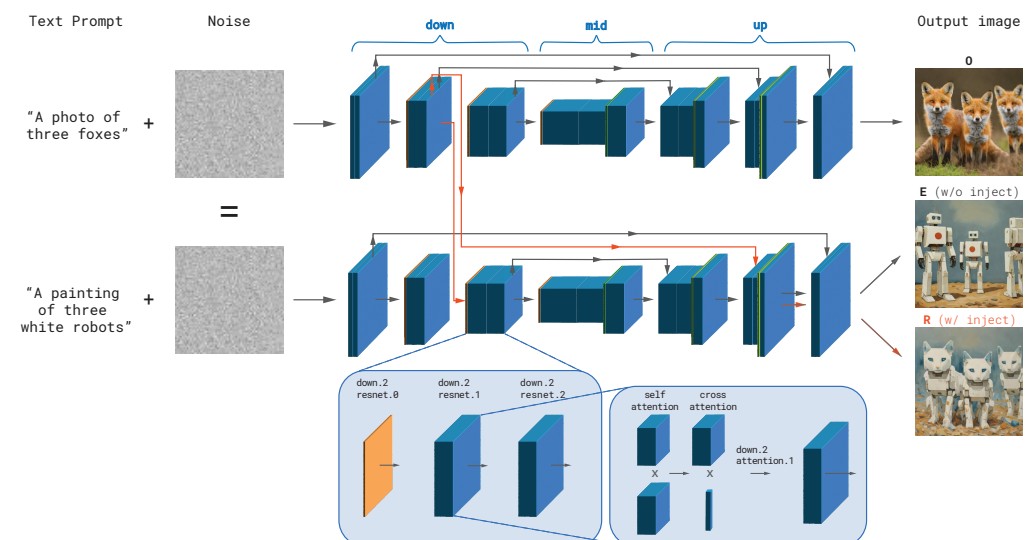

Figure 2: Injection process. Illustration of an example test carried out for the analysis. The selected layer from the denoising of image O is injected into the denoising of image E, leading to the resulting image R on the bottom right. The = indicates that both processes start from the same noise.

**Similarity Metrics.** We evaluate the resulting image $R$ on a scale from 0 (identical to $O$) to 1 (identical to $E$), thereby quantifying how strongly layer $i$ drives the image towards the edit. Four complementary descriptors are used: (i) SSIM for structural fidelity, (ii) SIFT keypoint overlap for layout consistency, (iii) color histogram similarity, and (iv) Local Binary Patterns (LBP) for texture. Together, these metrics disentangle semantic, structural, and stylistic influences (details in Appendix 7). We chose to rely solely on low-level descriptors for two primary reasons: i) feature descriptors such as CLIP (Ramesh et al., 2022) and LPIPS(Zhang et al., 2018) are often used in evaluation, and we wanted to ensure independence between the two steps; ii) the images are consistent from the point of view of shift, rotation, constrast and saturation, which makes them good candidates for quantification using low-level descriptors.

**Setup.** We test 729 combinations of 27 unique prompts E and O, each specifying a subject, style, and background. The 27 prompts are generated as follows, to include different combinations: "A high-resolution image of a {content}, in {background}, in the style of {style}", where: **content** = { zebra, parrot, elephant, desert } **background** = { desert, city, forest } **style** = { black and white, Japanese anime, neon colors }.

**Results.** Results for U-Net- and DiT-based backbones are provided in Figure 1. Our analyses reveal architecture-specific but interpretable representational signatures: U-Nets rely on bottleneck and early decoding stages, while DiTs concentrate mixing in deep transformer blocks. Several key patterns emerge. **Stable Diffusion 1.4 and 2** exhibit a strong peak at the late encoder and early decoder layers, while the bottleneck obtains image E without mixing (i.e., values around 0.5). SD1.4 shows a shorter spike during decoding compared to SD2. **Stable Diffusion XL** steadily accumulates editing strength through the encoder, peaking at the deepest block and bottleneck while still achieving mixing. **Kandinsky** diverges from SD models, with editing effects emerging late in the decoder stages. **Turbo models** closely track their teacher architectures, showing negligible representational differences. Across all U-Nets, down.2.resnets.1 emerges as a universally salient layer. **SD3.5** obtains mixing around middle/late transformer blocks 22–23. **Flux** concentrates on its final transformer block. Lastly, the color and texture mixing are generally lower for DiTs than U-Nets, compared to structural metrics.

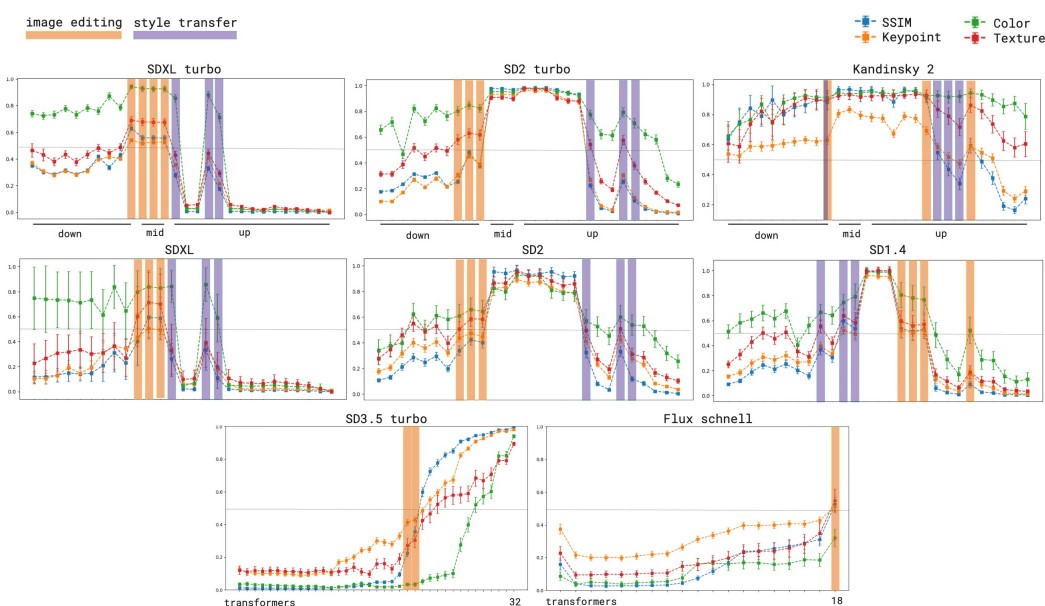

Figure 3: Visualization of the effect of each layer on a scale from the original image O - 0 - to the editing prompt E - 1. We only keep the common layers across all U-Net models for better comparison. The bars indicate 95% confidence intervals. As visible in the image, the models and turbo alternatives internally behave similarly, while the representations across models are largely different. We highlight the layers according to the explanation in Section 5.

## 5 FEATUREINJECT

We now introduce our editing and style transfer methodology, derived directly from the probing analyses.

**Method.** Our central observation is that effective editing occurs in layers whose injection effect lies *midway* between the original image $O$ and the edited prompt $E$, roughly around a similarity value of 0.5 in Figure 1. These 'mixing layers' represent points where structural and stylistic information from both trajectories overlap. We first identify all layers whose similarity values fall within the mixing region (in Figure 3). Then, we conduct ablation studies to test different subsets and combinations of these candidate layers. In fact, while the analysis assumes independence between layers for computational constraints, we overcome this assumption by testing different combinations of the candidate layers.

Two consistent patterns emerge. Layers where all four metrics (SSIM, keypoints, color, texture) converge around 0.5 yield balanced semantic editing (highlighted in orange in Figure 1). Layers where only chromatic and textural metrics diverge (highlighted in purple) primarily encode surface-level appearance and are best suited for style transfer. Our method, *FeatureInject*, selects and combines these layers accordingly: orange layers for content editing and purple layers for style transfer. For DiTs, following the observation that color and texture are never higher than structural metrics, we do not identify independent style transfer layers. Detailed ablation results validating these choices are provided in the Appendix 7.

**Datasets.** For evaluation, we follow established practice in text-guided image editing using the Wild-TI2I benchmark (Tumanyan et al., 2023). To additionally benchmark style transfer, we introduce **Wild-Style**, a companion dataset with identical structure and specifications but with edits restricted to stylistic attributes (dataset in Appendix 7). Each instance pairs an original prompt $O$ with an edited prompt $E$ describing only a target style. Styles span both historical artistic movements (e.g., Baroque, Impressionism, Cubism) and contemporary aesthetics (e.g., pixel art, neon

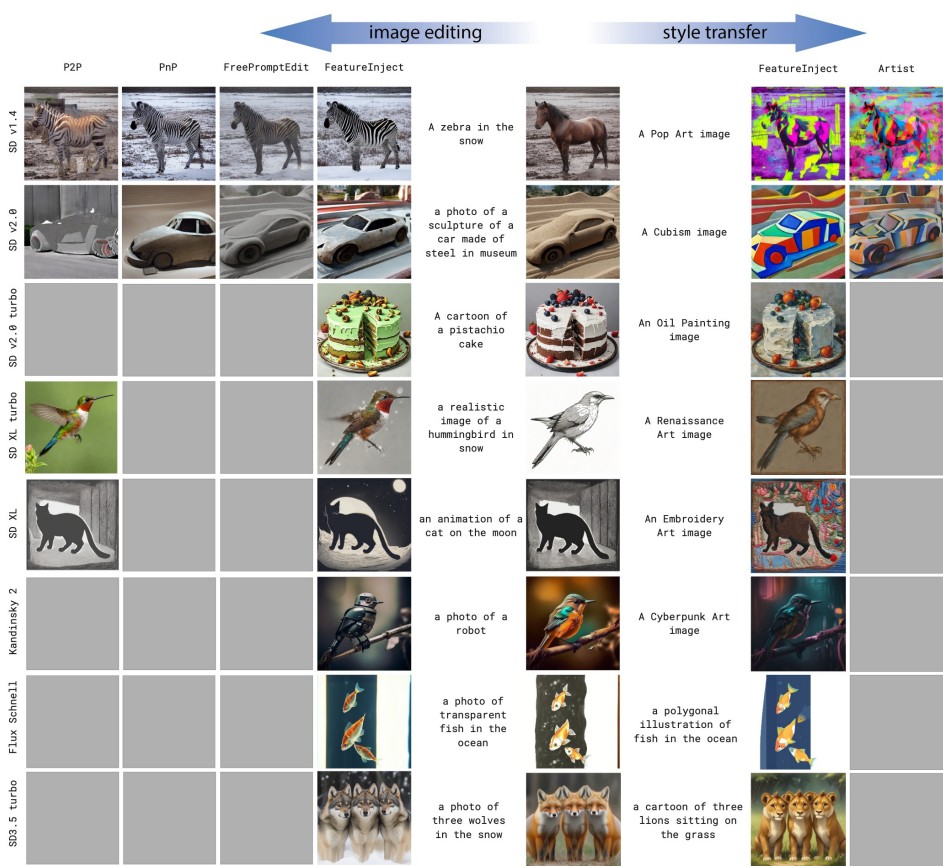

Figure 4: Comparison with existing methods for image editing, on the left; and style transfer, on the right. Whenever possible, we compare the results to SOTA image editing methods. The gray boxes correspond to applications of the methods that were not possible, to the best of our abilities. The results highlight both the incredible versatility of the method and its state-of-the-art editing results.

punk). This design ensures that evaluations cover both semantic content edits and surface-level style modifications.

**Metrics and Setup.** We adopt three complementary metrics to capture semantic alignment, structural fidelity, and perceptual similarity: (i) **CLIP similarity** measures how well generated images align with the target edit or style prompt (Radford et al., 2021); (ii) **Multi-scale SSIM** quantifies preservation of structural layout; (iii) **VGG feature similarity** evaluates semantic closeness to the image generated directly from the edited prompt.

All experiments are implemented with the `Diffusers` library. Feature injection is performed via PyTorch hooks, and images are generated with `AutoPipelineText2Image` using default parameters. For turbo-distilled variants, we follow prior work and use three inference steps. Details on model weights and the specific layers selected for injection are provided in the Appendix 7.

**Results.** Quantitative results are reported in Table 3, with qualitative comparisons in Figure 4. We benchmark against Artist (Jiang & Chen, 2024) for style transfer and Plug-n-Play (PnP) (Tumanyan et al., 2023), Prompt-to-Prompt (P2P) (Hertz et al.), and FreePromptEditing (FPE) (Liu et al., 2024) for image editing. When results are not available in Table 3 or Figure 4, it is because we could not, to the best of our abilities, adapt the implementations to generalize for that model/task. Therefore, we recognize the first major strength of our method: to generalize to the vastest set of models.

Furthermore, our method achieves a strong balance between prompt alignment and structure preservation. Compared to FPE, which often prioritizes structural fidelity at the cost of coherent edits, and

Table 3: Quantitative comparison of text-based image editing and style transfer across multiple diffusion models. Dashes (–) indicate methods or models could not be evaluated for that task. **Bold** indicates best performance for each model, while *italics* indicate best performance across models.

| Model | Method | Editing | | | Style Transfer | | |
|---|---|---|---|---|---|---|---|
| | | MS-SSIM ↑ | Feature Sim. ↑ | CLIP Sim. ↑ | MS-SSIM ↑ | Feature Sim. ↑ | CLIP Sim. ↑ |
| SD1.4 | P2P | 0.409 | 0.211 | 0.276 | – | – | – |
| | FPE | *0.625* | 0.231 | 0.273 | – | – | – |
| | PnP | 0.605 | 0.253 | **0.310** | – | – | – |
| | Artist | – | – | – | 0.174 | 0.133 | 0.175 |
| | Ours | **0.636** | **0.249** | 0.281 | **0.460** | **0.140** | **0.214** |
| SD2 | P2P | 0.297 | 0.194 | 0.266 | – | – | – |
| | FPE | 0.466 | 0.207 | 0.268 | – | – | – |
| | PnP | **0.559** | 0.260 | **0.332** | – | – | – |
| | Artist | – | – | – | 0.196 | 0.153 | 0.183 |
| | Ours | 0.550 | **0.267** | 0.306 | **0.428** | **0.180** | **0.250** |
| SD2-turbo | PnP | 0.336 | 0.173 | 0.174 | – | – | – |
| | Ours | **0.516** | **0.302** | **0.282** | 0.400 | 0.199 | 0.228 |
| SDXL-turbo | P2P | 0.283 | 0.362 | **0.297** | – | – | – |
| | Ours | **0.430** | **0.441** | 0.282 | 0.486 | 0.161 | 0.230 |
| SDXL | P2P | 0.386 | 0.099 | 0.290 | – | – | – |
| | Ours | **0.560** | **0.330** | **0.336** | *0.577* | 0.147 | 0.234 |
| Kandinsky | Ours | 0.403 | 0.385 | *0.368* | 0.383 | 0.272 | 0.295 |
| Flux Schnell | Ours | 0.491 | *0.480* | 0.274 | 0.426 | *0.520* | *0.300* |
| SD3.5 turbo | Ours | 0.412 | 0.250 | 0.238 | 0.362 | 0.220 | 0.266 |

P2P, which frequently introduces distortions, FeatureInject produces edits that are both consistent and semantically faithful. For style transfer, our approach yields visually coherent transformations, outperforming Artist in terms of completeness of stylistic change. Performance is consistently strong across SDXL, Kandinsky, and turbo variants, and importantly, generalizes effectively to DiT-based models such as Flux and SD3.5 Turbo.

## 6 OBSERVATIONS AND INTERPRETATIONS

Our analyses reveal distinct representational patterns across U-Net and DiT architectures. While causal mechanisms would require targeted ablations, we outline several plausible interpretations supported by prior work.

**Layer specialization.** Consistent with the information interpretation of gradient-descent-based learning as specialization for the task and compression of representation (Shwartz-Ziv & Tishby, 2017), we find that different layers encode different types of information due to their spatial resolution and embedding dimensionality. Early encoder and late decoder layers ($64 \times 64 \times 4$) emphasize fine-grained texture, while the bottleneck and deepest blocks ($1280 \times 8 \times 8$) capture global semantic composition. In line with prior studies (Voynov et al., 2023; Zhang et al., 2023; Agarwal et al., 2023), our results show that shallow layers mainly transmit structural fidelity, mid-layers capture semantic content, and early decoders strongly influence global style.

**Distribution of conditioning signals.** Stable Diffusion XL diverges from earlier models by concentrating 30–40 attention modules in its deepest encoder and shallowest decoder, while incorporating two separate text encoders. This shift in conditioning distribution may explain its steadily increasing mixing effect, though the precise causal role of dual conditioning remains open.

**Residual connections.** SD1.4 and SD2 display symmetric bell-shaped curves, with central layers almost entirely aligned to the editing prompt and outer layers converging to the same mixing level. This symmetry is consistent with skip-connection information flow, as also suggested by (Schaerf

et al., 2025; He et al., 2024). Direct probing of residual pathways would be needed to confirm this mechanism.

**Conditioning models and training data.** Despite identical architectures, SD1.4 and SD2 differ in representational dynamics. SD1.4 tends to establish style before content, whereas SD2 produces a more balanced progression. Since the main difference lies in conditioning (OpenAI CLIP vs. Open-CLIP trained on LAION-5B), this suggests that the training corpus can subtly shift representational allocation.

**Discrete vs. continuous decoding.** Kandinsky departs sharply from SD models. Its reliance on a MoVQGAN discrete decoder likely imposes higher representational demands on later decoder stages, explaining why mixing emerges primarily there. The interaction between discrete codebooks and diffusion dynamics remains an open question.

**Turbo distillation.** Distilled variants closely mirror their teacher models, but with sharper representational peaks. This supports the view that adversarial distillation preserves representational structure while concentrating information (Surkov et al., 2025). Distilled models may thus be more efficient but less flexible in how representations are distributed.

**Diffusion Transformers.** In contrast to U-Net backbones, Diffusion Transformers concentrate representational mixing in the final layers. We attribute this to the strictly sequential processing of transformer blocks, which lacks the long-range encoder–decoder connections of U-Nets. As a result, global interactions, driven by quadratic self-attention across patches and conditioning inputs, tend to accumulate gradually and manifest most strongly in the deepest layers. This design leads to more localized salient layers compared to the distributed representational roles observed in U-Nets.

# 7    CONCLUSION

We conducted a systematic, layer-wise analysis of diffusion backbones, probing representational structures across Stable Diffusion variants, SDXL, Kandinsky, turbo-distilled models, and two state-of-the-art DiTs. This led to several key insights. (i) Feature injection methods do not generalize uniformly: each architecture exhibits distinct representational regimes. (ii) Architectural choices, i.e., attention distribution, conditioning encoders, or decoder type, systematically influence representational allocation. (iii) Distillation preserves structure but amplifies activation peaks. (iv) DiTs achieve similar content–style–structure partitioning but with more localized layer responsibilities. Together, these findings argue against one-size-fits-all editing strategies and point to the need for model-specific probing when designing manipulation pipelines.

Building on these insights, we introduced *FeatureInject*, a principled editing framework that identifies and exploits 'mixing layers' for both semantic editing and style transfer. The method consistently achieves state-of-the-art performance across diverse models, being the first one to generalize to such a variety of models while maintaining structural fidelity and aligning closely with prompt semantics. Crucially, we demonstrate that the paradigm extends beyond U-Nets to DiT backbones, suggesting a general mechanism for training-free editing across diffusion architectures.

For practitioners, our probing methodology provides a systematic way to identify optimal injection layers in new models, reducing reliance on ad hoc trial-and-error. Our study highlights that architectural choices fundamentally shape representational structure in diffusion models. These insights open a path toward more robust, model-aware editing strategies and motivate future work on the theoretical understanding of representation formation in generative backbones. Future work should address the interaction between layers and modules more extensively at the level of the analysis.

## REPRODUCIBILITY STATEMENT

We added extensive detail regarding the implementation (specifying the library, the function used for injection, the names of the injected layers, the ablation studies for hyperparameters, and the links to the model weights). As stated, we used the default parameters for anything not specified. Furthermore, we specify the dataset used for evaluation and include the dataset introduced in the paper in Appendix 7. The repository containing the code will be released upon acceptance.

ETHICS STATEMENT

We disclose the use of LLMs in rephrasing, summarizing, or formatting parts of the text. Copilot was used during coding, with suggestions of minor length.

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

## APPENDIX A: LATENT DIFFUSION

Latent Diffusion Models (LDMs) address the computational and memory limitations of diffusion models by performing the diffusion process in a compressed latent space rather than the high-dimensional image space. Specifically, LDMs utilize a pre-trained autoencoder to encode images $x_0$ into a lower-dimensional latent representation $z_0 = E(x_0)$, where $E$ denotes the encoder. The diffusion process is then applied to $z_0$ instead of $x_0$, significantly reducing the computational resources required for training and sampling.

The forward diffusion process in the latent space is defined as:

$$z_t = \sqrt{\alpha_t} \cdot z_0 + \sqrt{1 - \alpha_t} \cdot \epsilon, \tag{3}$$

where $\epsilon \sim N(0, I)$ is Gaussian noise, and $\{\alpha_t\}$ represents a predefined noise schedule. The model learns to predict the added noise $\epsilon$ using a neural network $\epsilon_\theta(z_t, t)$ during training. The training objective minimizes the expected loss:

$$\mathcal{L}(\theta) = \mathbb{E}_{z_0,\epsilon,t} \left[ |\epsilon - \epsilon_\theta(z_t, t)|^2 \right], \tag{4}$$

where $t$ is uniformly sampled from the diffusion steps.

### COMPONENTS OF THE U-NET

This work focuses on pre-trained text-conditioned Latent Diffusion Models (LDM) with U-Net backbone, mainly Stable Diffusion versions 1.4, 2, 2.1 turbo, XL, XL turbo, and Kandinsky 2. We leave future work to assess the generalizability to pixel-level diffusion models such as Imagen (Ramesh et al., 2022) and DeepFloyd (Saharia et al., 2022), granted the computational resources. Given the general nature of the pipeline, we believe the extension is straightforward.

In Latent Diffusion, the traditional U-Net (Ronneberger et al., 2015) is augmented with attention modules - residual blocks, self-attention blocks, and cross-attention blocks—with cross-attention enabling text conditioning.

In particular, the **residual block** processes the latent features $\phi_t^{l-1}$ from the previous layer $l - 1$ to produce intermediate features $f_t^l$ and the latent features inputted to the following block $\phi_t^l$:

$$f_t^l, \phi_t^l = \text{ResBlock}(\phi_t^{l-1}), \tag{5}$$

where ResBlock includes convolutional operations and activation functions that capture local patterns in the data.

The **self-attention block** enables long-range interactions between latent features by computing attention scores among all spatial positions:

$$\hat{f}_t^l = \text{SelfAttn}(f_t^l) = A_t^l v_t^l, \tag{6}$$

where the attention matrix $A_t^l$ is computed using the queries $q_t^l$ and keys $k_t^l$:

$$A_t^l = \text{Softmax}\left(\frac{q_t^l k_t^{l\top}}{\sqrt{d_k}}\right), \tag{7}$$

with

$$q_t^l = W_q^l f_t^l, \quad k_t^l = W_k^l f_t^l, \quad v_t^l = W_v^l f_t^l. \tag{8}$$

Here, $W_q^l, W_k^l, W_v^l$ are learnable projection matrices, and $d_k$ is the dimensionality of the keys used for scaling.

The **cross-attention block** incorporates the text prompt $P$ into the image generation process. The text prompt is first embedded using a text encoder to obtain token embeddings $e = \text{TextEncoder}(P)$. The cross-attention between the latent features $f_t^l$ and the text embeddings $e$ is computed as:

$$\tilde{f}_t^l = \text{CrossAttn}(f_t^l, e) = \text{Softmax}\left(\frac{q_t^l e^\top}{\sqrt{d_k}}\right) e, \tag{9}$$

where $q_t^l = W_q^l f_t^l$ are the queries from the latent features, and $e$ serves as the keys and values.

## APPENDIX B: ANALYSIS SPECIFICATIONS

### METRICS

We compute four low-level image descriptors to quantify the amount of mixing between the original image O and the editing image E in the resulting image R. We assess the layout and content preservation using structural similarity (SSIM) and keypoint overlap (SIFT). Moreover, we consider color histogram similarity and texture similarity using Local Binary Pattern (LBP) for the style preservation.

For each macro layer in the U-Net - we do not consider the single multi-head attention modules - we compute the similarity to O and to E, as $SSIM_O$ between the original image $I_O$ and the resulting image with layer l changed $I_R(l)$, and $SSIM_E$ between $I_E$ and $I_R(l)$.

$$\text{SSIM}(I_1, I_2) = \frac{(2\mu_{I_1}\mu_{I_2} + C_1)(2\sigma_{I_1 I_2} + C_2)}{\left(\mu_{I_1}^2 + \mu_{I_2}^2 + C_1\right)\left(\sigma_{I_1}^2 + \sigma_{I_2}^2 + C_2\right)} \tag{10}$$

where $I_1$ and $I_2$ represent the grayscale images, $\mu_1$ and $\mu_1$ are their means, $\sigma_1$ and $\sigma_1$ their standard deviations, and $\sigma_{I_1}^2$ $\sigma_{I_1}^2$ their covariance. $C_1$ and $C_2$ are constants to stabilize the division.

To enrich the comparison on the layout, we use SIFT descriptors with K-Nearest Neighbor (KNN) matching and cosine similarity.

$$\text{Keypoint Similarity} = \frac{\sum_{i=1}^{n_1} \mathbb{I}\left(d_i^{(1)} < 0.75\, d_i^{(2)}\right)}{\max\left(|KP_1|, |KP_2|\right)}, \tag{11}$$

where $KP_1$ and $KP_2$ are the sets of SIFT keypoints detected in images 1 and 2, respectively, $n_1 = |KP_1|$ is the number of points found for image $I_1$, $d_i^{(1)}$ and $d_i^{(2)}$ are the smallest and second smallest Euclidean distances from the ith SIFT descriptor in image 1 to the descriptors in image 2 (obtained via KNN matching with $k = 2$), and $\mathbb{I}$ is the indicator function that equals 1 if the condition is true and 0 otherwise.

Finally, we measure features inherent to style as HSV color histogram similarities and Local Binary Patterns (LBP) histogram similarity for texture.

$$\text{Color / Texture Similarity} = \frac{\langle H_1 - \bar{H}_1, \, H_2 - \bar{H}_2 \rangle}{\|H_1 - \bar{H}_1\| \, \|H_2 - \bar{H}_2\|} \tag{12}$$

where $H$ is the HSV histogram and $\bar{H}$ the mean in the color similarity case, and LBP histogram for the texture.

Lastly, we normalize the similarity score by subtracting the similarity between the two input images, which is considered a baseline, and then normalizing their sum to 1.

$$\tilde{s}(O, R) = \frac{\text{sim}(O, R) - \text{sim}(O, E)}{\text{sim}(O, R) + \text{sim}(E, R)} \tag{13}$$

$$\tilde{s}(E, R) = \frac{\text{sim}(E, R) - \text{sim}(O, E)}{\text{sim}(O, R) + \text{sim}(E, R)} \tag{14}$$

| Model | Layers | MS-SSIM ↑ | Feature Sim. ↑ | CLIP Sim. ↑ |
|---|---|---|---|---|
| SD1.4 | up.0.res.0, up.0.res.1, up.0.res.2 | 0.3890 | 0.3295 | 0.3366 |
| SD1.4 | up.0.res.0, up.0.res.1 | **0.6356** | **0.2483** | **0.2813** |
| SD1.4 | up.0.res.2, up.0.res.0 | 0.3890 | 0.3295 | 0.3366 |
| SD1.4 | up.0.res.0 | 0.3964 | 0.3318 | 0.3398 |
| SD1.4 | up.0.res.0, up.0.res.1, up.0.res.2 | 0.6356 | 0.2483 | 0.2813 |
| SD1.4 | up.0.res.1, up.0.res.0 | 0.4082 | 0.3286 | 0.3359 |
| SD2 | down.2.res.0, down.2.att.1 | 0.4895 | 0.2992 | 0.3348 |
| SD2 | down.2.res.1, down.2.att.1 | 0.4737 | 0.2967 | 0.3314 |
| SD2 | down.2.res.0, down.2.att.1, down.2.res.1 | **0.5455** | **0.2669** | **0.3060** |
| SD2 | down.2.res.0, down.2.att.1, down.2.res.1 | 0.5177 | 0.2740 | 0.3129 |
| SDXL | mid.res.0 | 0.5141 | 0.3739 | 0.3628 |
| SDXL | mid.res.0, mid.res.1, down.2.res.1 | 0.5581 | 0.3290 | 0.3352 |
| SDXL | down.2.res.1 | 0.5139 | 0.3741 | 0.3634 |
| SDXL | mid.res.0, mid.res.1, down.2.res.1 | **0.5593** | **0.3283** | **0.3356** |
| SDXL | mid.res.0, mid.res.1 | 0.5033 | 0.3540 | 0.3529 |
| Kandinsky | up.1.res.0, down.2.res.1, up.0.res.2 | **0.4031** | **0.3852** | **0.3681** |
| Kandinsky | up.1.res.0, down.2.res.1, up.0.res.2 | 0.3909 | 0.3989 | 0.3697 |
| Kandinsky | down.2.res.1, up.0.res.2 | 0.3909 | 0.3989 | 0.3697 |
| Kandinsky | up.1.res.0, down.2.res.1 | 0.3203 | 0.5217 | 0.3763 |
| Kandinsky | up.0.res.2, up.1.res.0 | 0.3331 | 0.4569 | 0.3756 |
| SD2 turbo | down.2.res1 | 0.4311 | 0.3362 | 0.3094 |
| SD2 turbo | down.2.res0 | 0.4218 | 0.3372 | 0.3062 |
| SD2 turbo | down.2.att1 | **0.5164** | **0.3022** | **0.2817** |
| SD2 turbo | down.2.res1, down.2.res0 | 0.4782 | 0.3140 | 0.2895 |
| SDXL turbo | down.2.res1 | **0.4304** | **0.4407** | **0.3659** |
| SDXL turbo | mid.res0 | 0.4796 | 0.3745 | 0.3308 |
| SDXL turbo | mid.res1 | 0.4311 | 0.4390 | 0.3655 |
| SDXL turbo | down.2.res1, down.2.att1, mid.res0 | 0.4255 | 0.4402 | 0.3622 |
| SDXL turbo | down.2.res1, mid.res0, mid.res1, down.2.att1 | 0.4803 | 0.3749 | 0.3309 |
| SDXL turbo | mid.res0, mid.res1 | 0.3437 | 0.3179 | 0.3672 |
| SDXL turbo | down.2.res1, mid.res0, mid.res1 | 0.3368 | 0.3162 | 0.3709 |
| SD3.5 turbo | trans.23 | **0.412** | **0.2464** | **0.2383** |
| SD3.5 turbo | trans.22 | 0.4104 | 0.2309 | 0.213 |
| SD3.5 turbo | trans.22, trans.23 | 0.4114 | 0.2267 | 0.2131 |
| Flux Schnell | trans.18 | **0.4911** | **0.4797** | **0.2742** |

Table 4: Ablation study of image editing configurations. Layers follow compact notation: `down/up/mid.block.res/att-idx`.

## APPENDIX C: ABLATION STUDIES

Based on the layers highlighted in the comparative figure, we carry out an ablation study to determine the ideal combination of the identified layers. We use for all experiments injection timesteps [1000,200]. The results of these ablation studies can be found in Tables 4 and 5. The results show how the layers can be successfully identified using the analysis proposed in the main paper.

| Model | Layers | MS-SSIM ↑ | Feature Sim. ↑ | CLIP Sim. ↑ |
|---|---|---|---|---|
| SD1.4 | down.2.res0, mid.res1, mid.res0 | **0.4569** | **0.1403** | **0.2138** |
| SD1.4 | mid.res0 | 0.2864 | 0.2325 | 0.2774 |
| SD1.4 | down.2.res0, mid.res1 | 0.3067 | 0.1839 | 0.2404 |
| SD1.4 | down.2.res0 | 0.2528 | 0.2895 | 0.2781 |
| SD1.4 | down.2.res0, mid.res1 | 0.3800 | 0.1595 | 0.2265 |
| SD2 | up.1.res0, up.1.res1 | 0.3890 | 0.1994 | 0.2568 |
| SD2 | up.1.res0, up.1.res1 | **0.4275** | **0.1807** | **0.2477** |
| SD2 | up.1.res0, up.1.res1 | 0.5892 | 0.1301 | 0.1970 |
| SD2 | up.1.res1 | 0.5930 | 0.1309 | 0.1983 |
| SDXL | up.0.res0, up.0.att0 | **0.5769** | **0.1470** | **0.2341** |
| SDXL | up.0.res0, up.0.att0, up.0.res1 | 0.7686 | 0.1137 | 0.1693 |
| Kandinsky | up.1.att1, up.1.att0, up.1.att2 | **0.3827** | **0.2723** | **0.2945** |
| Kandinsky | up.1.att0, up.1.att1 | 0.3510 | 0.2858 | 0.3025 |
| Kandinsky | up.1.att0, up.1.att1 | 0.3311 | 0.2934 | 0.3046 |
| Kandinsky | up.1.att1 | 0.3510 | 0.2858 | 0.3025 |
| Kandinsky | up.1.att2, up.1.att1 | 0.3827 | 0.2723 | 0.2945 |
| SD2 turbo | up.1.res0, up.1.res1 | 0.4475 | 0.1685 | 0.2153 |
| SD2 turbo | up.1.res0, up.1.res1 | **0.4003** | **0.1991** | **0.2279** |
| SD2 turbo | up.1.res0, up.1.res1 | 0.6637 | 0.1073 | 0.1663 |
| SD2 turbo | up.1.res1 | 0.6637 | 0.1073 | 0.1663 |
| SDXL turbo | up.0.res0, up.0.att0 | **0.4855** | **0.1614** | **0.2303** |
| SDXL turbo | up.0.res0, up.0.att0, up.0.res1 | 0.6322 | 0.1081 | 0.1768 |
| SD3.5 turbo | trans.23 | **0.3623** | **0.2198** | **0.2664** |
| SD3.5 turbo | trans.22 | 0.3562 | 0.1349 | 0.1882 |
| SD3.5 turbo | trans.22, trans.23 | 0.3593 | 0.126 | 0.1822 |
| Flux Schnell | trans.18 | **0.4259** | **0.52** | **0.2981** |
| Flux Schnell | trans.17, trans.18 | 0.4514 | 0.2961 | 0.2136 |

Table 5: Ablation study of style transfer configurations. Layers follow compact notation: `down/up/mid.block.res/att-idx`.

## APPENDIX D: EVALUATION SPECIFICATIONS

### DATASETS

We evaluate the results using two datasets as explained in the main text. The first dataset, `wild-ti2i-fake.yaml` has been released by (Tumanyan et al., 2023), at the following link: Wild-ti2i. We chose this dataset as it is widely used in the literature and it tests a variety of editing prompts, specifying the seed for randomness, DDIM (Song et al.) steps and classifier free guidance (Ho & Salimans). As real image editing strongly depends on the inversion technique, we chose to test only on the 'fake' dataset of generated image editing. We further introduce a second dataset `wild-style-fake.yaml` that mirrors all the specs of the previous, but the editing is focused on style transfer only. The full dataset is shown in the **Style Dataset** section.

### METRICS

As explained in the main text, we use three metrics to evaluate the results. We choose CLIP to compare the text of the editing to the generated image, as standard in the field. We load CLIP using OpenCLIP 'ViT-B-32' trained on laion2b_s34b_b79k. To evaluate the content of the edited image

with the image generated by the editing prompt we use VGG16 from `torchvision`. Lastly, we compute the multiscale structural similarity to assess structural coherence between the original image and the edited image.

In the main text, when methods are not available, it is because the code released and rationale for these methods do not natively work on those models, and the authors could not find other implementations nor implement them themselves without changing the rationale.

Lastly, we choose the best performing model in the ablation based on a tradeoff between the structural similarity and the VGG-CLIP similarities.

STYLE DATASET

```
1   # horse in mud
2   - source_prompt: a photo of a horse in mud
3     seed: 50
4     scale: 5.0
5     ddim_steps: 50
6     target_prompts:
7        - A Renaissance painting image
8        - A Cubism image
9        - A Surrealism image
10       - A Pop Art image
11
12  # Spider man
13  - source_prompt: a photo of spider man in superhero pose
14    seed: 90
15    scale: 5.0
16    ddim_steps: 50
17    target_prompts:
18       - An Impressionism image
19       - A Baroque style image
20       - A Futurism image
21
22  # Two cats
23  - source_prompt: a photo of two cats in the garden
24    seed: 50
25    scale: 7.5
26    ddim_steps: 50
27    target_prompts:
28       - A Dadaism image
29       - An Art Nouveau image
30       - A Minimalism image
31
32  # White tesla
33  - source_prompt: a photo of a white Tesla
34    seed: 62
35    scale: 7.5
36    ddim_steps: 50
37    target_prompts:
38       - A Chinese Ink image
39       - An Embroidery Art image
40       - An Oil Painting image
41
42  # three foxes
43  - source_prompt: a photo of three foxes
44    seed: 123
45    scale: 7.5
46    ddim_steps: 50
47    target_prompts:
48       - A Watercolor Painting image
49       - A Studio Ghibli image
50       - A Cyberpunk image
```

```
 51
 52   # christmas trees
 53   - source_prompt: a photo of christmas trees above snow
 54     seed: 3000
 55     scale: 7.5
 56     ddim_steps: 50
 57     target_prompts:
 58       - A Pixel Punk image
 59       - A Wasteland image
 60       - A Sketching image
 61       - A Renaissance painting image
 62
 63   # bunny doll
 64   - source_prompt: a photo of a blue bunny doll
 65     seed: 1010
 66     scale: 7.5
 67     ddim_steps: 50
 68     target_prompts:
 69       - A Cubism image
 70       - A Surrealism image
 71       - A Pop Art image
 72       - An Impressionism image
 73
 74   # white bird
 75   - source_prompt: a photo of a white bird on the grass
 76     seed: 1010
 77     scale: 7.5
 78     ddim_steps: 50
 79     target_prompts:
 80       - A Baroque style image
 81       - A Futurism image
 82       - A Dadaism image
 83       - An Art Nouveau image
 84
 85   # green ducks
 86   - source_prompt: a photo of green ducks walking on street
 87     seed: 7000
 88     scale: 7.5
 89     ddim_steps: 50
 90     target_prompts:
 91       - A Minimalism image
 92       - A Chinese Ink image
 93       - An Embroidery Art image
 94
 95   # Cake
 96   - source_prompt: a photo of a cake
 97     seed: 30
 98     scale: 7.5
 99     ddim_steps: 50
100     target_prompts:
101       - An Oil Painting image
102       - A Watercolor Painting image
103       - A Studio Ghibli image
104
105   # songbird
106   - source_prompt: a beautiful image of a song bird
107     seed: 77
108     scale: 7.5
109     ddim_steps: 50
110     target_prompts:
111       - A Cyberpunk image
112       - A Pixel Punk image
113       - A Wasteland image
114
115   # sand cars
```

```
116  - source_prompt: a photo of a car made of sand
117    seed: 800
118    scale: 7.5
119    ddim_steps: 50
120    target_prompts:
121      - A Sketching image
122      - A Renaissance painting image
123      - A Cubism image
124
125  - source_prompt: a photo of a car made of sand
126    seed: 3003
127    scale: 7.5
128    ddim_steps: 50
129    target_prompts:
130      - A Surrealism image
131      - A Pop Art image
132      - A Impressionism image
133
134  # black bycicle
135  - source_prompt: a photo of a black bicycle
136    seed: 9090
137    scale: 5.0
138    ddim_steps: 50
139    target_prompts:
140      - A Baroque style image
141      - A Futurism image
142      - A Dadaism image
143
144  # cat face
145  - source_prompt: a photo of a cat face
146    seed: 500
147    scale: 7.5
148    ddim_steps: 50
149    target_prompts:
150      - An Art Nouveau image
151      - A Minimalism image
152      - A Chinese Ink image
153
154  # mouse painting
155  - source_prompt: a painting of a gray mouse
156    seed: 12
157    scale: 7.5
158    ddim_steps: 50
159    target_prompts:
160      - An Embroidery Art image
161      - An Oil Painting image
162      - A Watercolor Painting image
163
164  # fish drawing
165  - source_prompt: a drawing of goldfishes in the ocean
166    seed: 500
167    scale: 5.0
168    ddim_steps: 50
169    target_prompts:
170      - A Studio Ghibli image
171      - A Cyberpunk image
172      - A Pixel Punk image
173
174  # Minimal bird
175  - source_prompt: a minimal drawing of a bird
176    seed: 77
177    scale: 7.5
178    ddim_steps: 50
179    target_prompts:
180      - A Wasteland image
```

```
181        – A Sketching image
182        – A Renaissance painting image
183        – A Cubism image
184
185   # blue car cartoon
186   – source_prompt: a cartoon of a blue car
187     seed: 310
188     scale: 7.5
189     ddim_steps: 50
190     target_prompts:
191        – A Surrealism image
192        – A Pop Art image
193        – An Impressionism image
194
195   # sketch mountain
196   – source_prompt: a sketch of a mountain
197     seed: 310
198     scale: 7.5
199     ddim_steps: 50
200     target_prompts:
201        – A Baroque style image
202        – A Futurism image
203        – A Dadaism image
204
205   # silhouette cat
206   – source_prompt: a silhouette drawing of a cat
207     seed: 610
208     scale: 7.5
209     ddim_steps: 50
210     target_prompts:
211        – An Art Nouveau image
212        – A Minimalism image
213        – A Chinese Ink image
214        – An Embroidery Art image
215        – An Oil Painting image
```

