# OpenReview forum: "One Size Does not Fit All: Cross-Architectural Layer-wise Representations in Diffusion Models"
_ICLR.cc/2026/Conference — ICLR 2026 Conference Withdrawn Submission_

### Official Review · Reviewer_Qoe3 · 2025-10-24

**Soundness:** 2
**Presentation:** 2
**Contribution:** 1
**Rating:** 2
**Confidence:** 4

**Summary:**

This paper presents a study of how individual layers in diffusion models form representations that enable image editing or style transfer. It localizes crucial layers for these tasks by injecting internal features from a denoising process conditioned on an edited prompt into the process for an original prompt. Based on this analysis, the paper forms observations about the nature of representations across layers in different text-to-image diffusion model architectures.

**Strengths:**

- The paper is clearly written.
- The insights into the functions of layers across different diffusion models are useful and can contribute to better model interpretability.

**Weaknesses:**

- The details on layer selection presented in Figure 3 are confusing. In line 310 it is stated that layers selected for image editing score near 0.5 on all metrics. However, in the provided examples, such as for SDXL turbo, the layers selected for editing score nearly 1.0 on the color metric. It is also unclear what interval is assumed to be "close to 0.5".
- It is difficult to evaluate the true performance of the method for DiT-based models without any baselines. While P2P may not be straightforwardly applicable to DiTs, could the method be adapted to serve as a baseline, for instance by following the approach in [1] where only specific attention components are edited?
- The observations and methodology are similar to Stable Flow [2], with the distinction that Stable Flow identifies vital layers by removing them, not injecting activations. The paper should compare to this method and elaborate on the differences.
- The analysis relies on injecting features from only a single layer. While this can surface insights, it is not obvious how it will scale to future models where layers may not specialize as cleanly. For instance, in the Flux Schnell results, only one layer achieves an editing score close to 0.5. It is possible that in future models, it will be impossible to localize this capability to a single layer.
- Figure 4 presents a very limited number of qualitative samples. Displaying results for a different prompt for each model also raises concerns that the examples may be cherry-picked.
- The FeatureInject method is fundamentally a simple activation patching technique, which is well-established in the literature. [3]

[1] Staniszewski, Łukasz, et al. "Precise Parameter Localization for Textual Generation in Diffusion Models." _The Thirteenth International Conference on Learning Representations_.

[2] Avrahami, Omri, et al. "Stable flow: Vital layers for training-free image editing." _Proceedings of the Computer Vision and Pattern Recognition Conference_. 2025.

[3] Meng, Kevin, et al. "Locating and editing factual associations in gpt." _Advances in neural information processing systems_ 35 (2022): 17359-17372.

**Questions:**

- Could a simple prompt-based baseline be used to verify the method's effectiveness, especially in the style transfer evaluation? For example, for the "zebra in the snow" to "pop art image" task, the prompt could be "A pop art image of a zebra in the snow."
- Can you compare your method to adapted P2P and Stable Flow. This would increase the reliability of evaluation results.
- Could you provide more editing examples, including failure cases?

---

### Official Review · Reviewer_ZUko · 2025-10-30

**Soundness:** 3
**Presentation:** 2
**Contribution:** 3
**Rating:** 4
**Confidence:** 4

**Summary:**

This paper conducts a layer-wise analysis about how semantic and stylistic information propagates through U-Net and transformer backbones. There are several diffusion models with variant architectures are evaluated. The best injection layers are selected based on the interactions between four low-level metrics. The authors identify that different architectures behave differently in terms of how the layers propagate the information.

**Strengths:**

1. It’s interesting to see a systematic evaluation of the layer-wise feature injection for different diffusion backbones. It’s true that most of the existing editing methods are usually only tested for very specific architectures. It’s good to see a performance analysis across different models in one paper.

2. The proposed method to find the best injection layer is straightforward and intuitive.

**Weaknesses:**

1. The selection of similarity metrics seems to be very important to the final analysis and editing results. I didn’t find a justification for why those 4 metrics are selected (If all of them are necessary? If any other metrics are possible?) Also, despite the high-level metrics like CLIP and LPIPS being commonly used in the evaluation, they may also serve as meaningful metrics, complemented by other metrics, to find the best injection layers.

2. For the qualitative comparison in Figure 4, I think there is not enough comparison between the proposed method and the baselines, especially for style transfer, which I believe there are more text-guided style transfer methods available. I also believe that there are more editing methods designed for DiT architectures that are stronger than the P2P, PnP, and FreePromptEdit selected in the paper (for examples, Flux Kontext). Right now, the qualitative comparison is not strong and consistent enough according to Figure 4.

3. Suggestion: In my opinion, if many model variants for the existing methods are not possible, the authors can only provide the results for the model variants that are possible for most of the methods in the main paper, and therefore they can also put more editing examples, and maybe more baseline methods.

**Questions:**

1. Does it mean that for transformers-based architectures, simple feature injection cannot achieve style transfer?

2. Could the authors provide visual examples on how the edited results are like for the effect in different layers? Basically, I’m curious about a visual comparison for a given model in Figure 3, so I can know how the authors determine that those layers are best suited for semantic editing / style transfer beyond the metrics ablation in appendix C. The authors can pick any model and a few editing examples to demonstrate.

3. Can the authors point out the limitations of the layer-wise feature injection method in general? For example, are edits beyond semantic editing and style transfer possible, or layer-wise injection methods are inherently not suitable for edits like deformation?

In general, I'm willing to update my rating if the authors can solve my concerns in the weaknesses section and answer the questions here.

---

### Official Review · Reviewer_cD91 · 2025-11-01

**Soundness:** 3
**Presentation:** 3
**Contribution:** 2
**Rating:** 2
**Confidence:** 3

**Summary:**

Researchers have developed a lot of training-free editing techniques that directly manipulate model features. However, those methods require layer searching and cannot be freely applied with the recently released base model (e.g., Flux). The author provides investigations of existing methods and finds optimal layers for them.

**Strengths:**

- Thorough investigation across multiple baselines, including SD1.4, SD2, SD2-turbo, SDXL, Kandinsky, Flux Schnell, and SD3.5 turbo.

**Weaknesses:**

While the paper presents an interesting direction, the contribution appears somewhat incremental relative to prior works.
- Feature injection techniques have been previously explored in diffusion-based image editing and style transfer [StyleInjection, Injectfusion]
- Layer selection for image editing and style transfer has also been studied in earlier methods, which investigate how manipulating attention or feature layers can improve controllability or consistency [Stylekeeper, MasaCtrl, Asyrp, InstantStyle]

[style injection]: Style Injection in Diffusion: A Training-free Approach for Adapting Large-scale Diffusion Models for Style Transfer
[injectfusion]: Training-free Content Injection using h-space in Diffusion Models
[StyleKeeper]Prevent Content Leakage using Negative Visual Query Guidance
[MasaCtrl]: Tuning-Free Mutual Self-Attention Control for Consistent Image Synthesis and Editing
[Asyrp]: Diffusion models already have a semantic latent space
[InstantStyle&InstantStyle2]: Free Lunch towards Style-Preserving in Text-to-Image Generation

**Questions:**

- Why do different architectures or models show different characteristics in their layers?
- Based on the analysis presented in the paper, is it possible to predict what kinds of properties future models might have or to generalize these findings?

---

### Official Review · Reviewer_m5Cp · 2025-11-04

**Soundness:** 2
**Presentation:** 2
**Contribution:** 1
**Rating:** 2
**Confidence:** 4

**Summary:**

This submission evaluates existing methods for localization of the knowledge in diffusion models on several different models showing differences between localization in UNets vs. Transformers, including their distilled version. On top of that, authors evaluate that edits performed with only the selected layers are often better than simple benchmarks like full prompt to prompt method.

**Strengths:**

This work is well motivated. The tackled problem which is a consistency of features localization in different diffusion models is a challenging and important issue that might be useful in numerous applications and have a significant impact.

**Weaknesses:**

- “We introduce a probing-based methodology that quantifies the representational role of each layer using structural and stylistic similarity metrics.” - Introduced methodology was already used by multiple papers. Apart from those mentioned in the submission see for example:

[1] Meng, Kevin, et al. "Locating and editing factual associations in gpt." Advances in neural information processing systems 35 (2022): 17359-17372.

[2] Zarei, Arman, et al. "Localizing Knowledge in Diffusion Transformers." arXiv preprint arXiv:2505.18832 (2025).

[3] Basu, Samyadeep, et al. "On mechanistic knowledge localization in text-to-image generative models." Forty-first International Conference on Machine Learning. 2024.

[4] Staniszewski, Łukasz, et al. "Precise Parameter Localization for Textual Generation in Diffusion Models." International Conference on Learning Representations 2025.

[5] Avrahami, Omri, et al. "Stable flow: Vital layers for training-free image editing." Proceedings of the Computer Vision and Pattern Recognition Conference. 2025.

	In its current form the claim seems to be an overstatement, what reduces the novelty of the proposed research

- Given the fact that this submission rather reuses existing techniques to apply them on different models, therefore treating this research more as survey, the evaluation with 4 objects with 3 different backgrounds and in 3 styles is not sufficient for drawing any general conclusions. This is also visible in Figure 4, where the evaluation is clearly incomplete with several combinations missing.

- Edition performance is compared (in most of the cases) only with extremely simple P2P baseline. More recent state-of-the-art approaches like the one mentioned above are missing

**Questions:**

-

---

### Note · Authors · 2025-12-03

I have read and agree with the venue's withdrawal policy on behalf of myself and my co-authors.